# Evaluating dyadic factors associated with self-care in patients with heart failure and their family caregivers: Using an Actor-Partner Interdependence Model

JinShil Kim[1], Kye Hun Kim[2], Mi-Seung Shin[3], Seongkum Heo[4], Jung-Ah Lee[5], KyungAh Cho[1], Minjeong An[6]*

1 College of Nursing, Gachon University, Incheon, South Korea, 2 Department of Cardiovascular Medicine, Medical School, Chonnam National University, Gwangju, South Korea, 3 Gil Medical Center, Division of Cardiology, Department of Internal Medicine, College of Medicine, Gachon University, Incheon, South Korea, 4 Georgia Baptist College of Nursing, Mercer University, Atlanta, GA, United States of America, 5 Sue & Bill Gross School of Nursing, University of California, Irvine, Irvine, CA, United States of America, 6 College of Nursing, Chonnam National University, Gwangju, South Korea

* anminjeong@jnu.ac.kr

**Data Availability Statement:** All relevant data are within the paper and Supporting Information files.

## Abstract

Dyadic conditions of patients with heart failure and their caregivers may affect both patient self-care and caregiver contribution to patient self-care (CCPS). The purpose of this study was to examine the relationships of patient-caregiver physical function and depressive symptoms to the patient self-care (maintenance and management) and CCPS. Data from 55 were analyzed using an Actor–Partner Interdependence Model to address the aim through AMOS. Patient self-care was very poor. Better patient physical function was related to better patient self-care management (actor effect) and poorer CCPS maintenance (partner effect). Better caregiver physical function was related to CCPS management (actor effect). Severer patient depressive symptoms were related to poorer patient self-care maintenance (actor effect) and poorer CCPS management (partner effect). Physical function and depressive symptoms in patient-caregiver dyads were related to patient self-care and CCPS. To improve patient self-care and CCPS, dyadic support for physical function and depressive symptoms is needed.

## Introduction

Heart failure (HF) is a chronic debilitating condition, which commonly results in poor quality of life [1–3] and high rates of hospitalization and mortality [4, 5]. Because self-care affects the status of these health outcomes [1–4], patients with HF need life-long efforts to adhere to the therapeutic guidelines, which is self-care. However, self-care in patients with HF is considerably poor [6–9]. Thus, self-care in patients with HF needs to be improved.

One important factor affecting patient self-care is social support from significant others, including family caregivers, because patients with HF need to adhere to a wide spectrum of

**Funding:** The National Research Foundation of Korea for Jane Kim supported part of this work for JinShil Kim (Grant #: NRF-2021R1A2C2009491). URL: http://nrf.re.kr The funders had no role in study design, data collection and analysis, decision to publish, or preparation of the manuscript.

**Competing interests:** The authors have declared that no competing interests exist.

treatment regimens [10, 11]. In patients with HF, a caregiver can support and assist the patient to understand and follow complex therapeutic recommendations to maintain the body stability (self-care maintenance), recognize signs and symptoms (symptom perception), and to manage symptoms when they occur (self-care management) [12–16]. A caregiver can contribute to the patient's self-care maintenance by reminding and encouraging the patient to take medication as prescribed, encouraging the patient to perform regular physical activities and doing them with the patient, and preparing for and providing low sodium foods to the patient [13]. A caregiver can contribute to the patient's symptom recognition by monitoring and recognizing signs and symptoms of patients [13, 16]. A caregiver can contribute to the patient's self-care management by assisting the patient to determine and manage the causes of signs and symptoms [13, 16]. Thus, in the literature, caregivers' active involvement in patient self-care improved patient self-care and reduced patient hospitalization rates [12, 17–20]. Therefore, patient-caregiver dyad needs to be considered to improve patient self-care and health outcomes.

Both in patients with HF and in their caregivers, depressive symptoms are prevalent and a factor affecting both patient self-care and caregiver contribution to patient self-care (CCPS). In patients with HF, the prevalence of depressive symptoms ranged from 32% to 44% [21, 22], in caregivers of patients with HF, the prevalence ranged from 6% to 64% [23–25]. In another study [26], approximately 21% of both patients with HF and their caregivers had depressive symptoms. High prevalence of depressive symptoms in patients may be due to the debilitating nature of HF symptoms [27]. The prevalence of depressive symptoms in caregivers may be due to increase in caregiving burden [25, 28], which is increased with becoming more severe HF symptoms in patients [29]. In patients with HF, depressive symptoms were consistently associated with poor self-care in a systematic literature review [30] and in caregivers of HF patients, the absence of depressive symptoms or anxiety was associated with an improved quality of life for the patients [31]. In addition, depressive symptoms in patients and in caregivers were associated with increased caregiver burden [28, 32], which may impact patient self-care [33, 34]. Therefore, depressive symptoms in patients with HF and in their caregivers can negatively affect patient self-care and CCPS. However, these relationships in patient-caregiver dyad have not been frequently examined.

Another factor possibly affecting both patient self-care and CCPS is physical functional status of both patients and their caregivers. Physical function in patients with HF is considerably impaired due to HF and HF symptoms [35–37]. Impaired physical function in patients with HF can positively or negatively affect patient self-care [38–40]. For example, patients with impaired physical function due to development or worsening of HF symptoms may feel urgent needs to be involved in self-care to reduce HF symptoms. However, in some cases, patients with impaired physical function may not be involved in self-care actively because of limited physical capacities in preparing and following a low sodium diet and performing regular exercise. Patients with impaired physical function can also affect CCPS because of increased demands and time for caregiving [41].

On the other hand, physical function in caregivers of patients with HF could be impaired because of lack of time and effort for their own self-management due to caregiving [42, 43]. Impaired physical function of caregivers can adversely affect CCPS because of their limited capacity for shopping and preparing low sodium foods for patients, and of recognizing of patients' signs and symptoms and supporting symptom management, which will also adversely affect patient self-care. The relationships of physical function of caregivers to CCPS or patient self-care have been rarely examined in HF. In one study, physical aspects of quality of life assessing the effects of physical function on daily lives were significantly associated with patient self-care management [44]. In addition, the relationships of physical function of

patients and caregivers to patient self-care also have not been frequently examined in HF. Therefore, further research is needed to examine those relationships in patients with HF and in their caregivers.

The purpose of this study was to examine the relationships of depressive symptoms and physical function to CCPS (i.e., contribution to patient self-care maintenance and contribution to patient self-care management) and patient self-care (self-care maintenance and self-care management) in patient-caregiver dyads using an Actor-Partner-Interdependence Model (APIM). An APIM is a useful method to promote understanding of the dyadic experiences and their interrelated characteristics [31, 45]. Specifically, we aimed to (1) compare the levels of dyadic physical function, depressive symptoms, and self-care in HF, and (2) examine whether physical function and depressive symptoms in patient-caregiver dyads were associated with patient self-care and CCPS. We hypothesized that physical function and depressive symptoms in patients with HF and in their caregivers as dyads would affect patient self-care and CCPS, respectively (i.e., actor effects), and their partner's outcome (i.e., partner effects), respectively.

## Methods

### Study design and setting

A cross-sectional, correlational study design was used to examine the physical function and depressive symptom status of patients with HF and their caregivers as dyads, and their actor and partner associations with patient self-care and CCPS using the data of a parent study. The purpose of the parent study was to examine patient and caregiver factors related to different types of self-care and self-care confidence in patients with HF using non-dyadic analysis methods [46]. The dyadic data of patients and their caregivers for this secondary analysis study were obtained from four university-affiliated hospitals in South Korea. Physicians and nurses at each hospital screened and enrolled patients with HF and their caregivers when they visited hospitals from June 12, 2019 to October 29, 2021.

### Sample

Patients were eligible to participate in this study if the following criteria were met: adult patients ($\geq$ 19 years) with a diagnosis of HF and prescription of any HF medication(s) based on the professional guidelines (e.g., beta-blockers, angiotensin-converting enzyme inhibitors, angiotensin receptor blockers, and diuretics) [47]. Caregivers of patients were eligible if the following criteria were met: adults ($\geq$ 19 years), an informal primary caregiver of a patient based on self-report, and no diagnosis of HF. Communal exclusion criteria for both patients and caregivers were end-stage of HF, terminal comorbid conditions (e.g., cancer or organ failure with life expectancy < 6 months), or no documented or self-reported neurocognitive disorders (e.g., stroke, Alzheimer, dementia, or memory disorder, or mental disorders), which could possibly accompany cognitive impairment and preclude understanding of and following the recommended guidelines to manage the care of HF. The judgement of these exclusion criteria was done by the physicians and nurses at each hospital who screened and recruited patients through reviews of the electronic medical records.

For this dyadic analysis, APIM with distinguishable dyads was used [48]. A minimum sample size was estimated to detect the actor and partner effects, given a power of .80 and an alpha of .05. In a prior study [49], 41 patient-caregiver dyads were estimated to examine the effects of patients' and caregivers' emotional distress (independent variables) on their own quality of life as well as their partner's (dependent variable) using APIM regression for distinguishable dyads [49]. Considering the prior study [49], 49 patient-caregiver dyads were needed to have

adequate power (80%) to detect the average effect sizes of the Person 1 and Person 2 actor effects (.275, .250). In this study, data from 55 patient-caregiver dyad were analyzed to address the purpose of this study.

## Measurements

The patient and caregiver dyads provided data on patient self-care (patients only), CCPS (caregivers only), physical function (patients and caregivers), and depressive symptoms (patients and caregivers).

**Patient self-care.** Patient self-care was assessed by the Revised Self-Care of Heart Failure Index, v7.2 (the revised SCHFI) [1]. The revised SCHFI, v7.2 consists of 29 items in the three scales, including the Self-care Maintenance (10 items), the Symptom Perception (11 items), and the Self-care Management (8 items) [1]. For this dyadic analysis, the Self-care Maintenance and Self-care Management scales were used. Each scale of the revised SCHFI basically adopts a five-point Likert scale. Following the scoring algorithm, 0 to 100 standardized scores for each of the scales are computed [50], with higher scores indicating higher levels of self-care maintenance and self-care management, respectively. A cut-point of 70 or greater in each scale indicates as adequate self-care [1]. The reliability and validity of the original version [1, 3] and of the Korean version [51] were supported.

**Caregiver contribution to patient self-care.** Caregiver contribution to patient self-care was assessed by the Caregiver Contribution to Self-care of Heart Failure Index, v1.0 [CC-SCHFI]) [52]. The CC-SCHFI consists of 22 items in the three scales, including the Caregiver Contribution to Self-care Maintenance (10 items), the Caregiver Contribution to Self-care Management (6 items), and the Caregiver Contribution to Self-Care Self-Efficacy (6 items). For this dyadic study, the Caregiver Contribution to Self-care Maintenance and the Caregiver Contribution to Self-care Management scales were used. One item in the CC-SCHFI Self-care Management assessing recognition of patient symptoms when they present was excluded with the approval from the developer in order to include all caregivers of patients with and without HF symptoms. Each scale of the CC-SCHFI basically adopts a four-point Likert scale. Following the scoring algorithm, 0 to 100 standardized scores for each scale was computed [50, 52], with higher scores indicating higher levels of caregiver contribution to self-care maintenance and caregiver contribution to self-care management, respectively. A cut-point of 70 or greater in each scale is assumed as adequate CCPS [1]. The reliability and validity of the original version were supported [53]. The Korean version of the CC-SCHFI was approved by the developer and posted on the website [54], while its psychometric testing is undertaking by authors of this study.

**Physical function.** Physical function was assessed by the Korean Activity Status Index (KASI) in patients with HF and their caregivers [55]. The KASI consists of 15 daily activities, which has a weighted value for the energy expenditure of each activity [55]. Using a dichotomous option (yes or no), participants are asked whether they were able to perform these activities. A total score is a sum of these weighted values. The possible total scores range from 0 to 79, with higher scores indicating higher levels of physical function. The validity of the KASI was supported by a moderate correlation coefficient between the KASI score and treadmill exercise time (rs = 0.62) and 77% agreement between the KASI-estimated functional class and functional class estimated by exercise [55].

**Depressive symptoms.** Depressive symptoms were assessed by the Patient Health Questionnaire-9 (PHQ-9) in patients with HF and in their caregivers [56, 57]. Each item adopts a four-point Likert scale (from 0 = *never* to 3 = *always*). The possible total scores range from 0 to 27, with higher scores indicating more severe depressive symptoms. The reliability and validity

of the English version in patients with HF and the Korean version in a geriatric population have been supported [57, 58].

**Demographic characteristics.** Demographic characteristics of both groups were collected using a standard demographic questionnaire. The demographic characteristics included age, sex, marital status, education level, and the patient-caregiver relationship.

**Clinical characteristics.** Clinical characteristics of patients were collected by the trained research nurses at each of the four sites using a standardized data abstraction form through medical record reviews. Clinical characteristics included etiology of HF, left ventricular ejection fraction, and prescribed medications. The New York Heart Association (NYHA) functional class of patients was determined by the trained nurses. Caregivers also self-reported if they had any diseases.

### Data analysis

Data were analyzed using IBM SPSS Statistics for Windows [59] and AMOS 27.0 software [60]. Patient and caregiver data were paired before data analyses and a significance level of < .05 was used in all tests. Descriptive statistics, such as mean with standard deviation or frequency with percent, were performed to describe the sample characteristics of patient and caregiver dyads and research variables. Independent t tests and $\chi 2$ tests were conducted to compare the differences in continuous and categorical sample characteristics between patients and caregivers, respectively. Independent t tests or Mann-Whitney U tests were used to compare the differences in physical function, depressive symptoms, and self-care maintenance and management between patients and caregivers depending on the normality in the distribution of each variable.

A structural equation modeling (SEM) approach was employed to estimate the APIM with distinguishable dyads. In this method, actor effects (associations between an individual's physical function or depressive symptoms and his or her own self-care maintenance and management or contributions) and partner effects (associations between an individual's physical function or depressive symptoms and his or her partner's self-care maintenance and management or contributions). Both actor and partner effects of physical function and depressive symptoms were examined as predictors of patient self-care or CCPS. To report standardized coefficients as well as unstandardized coefficients, second SEM was conducted after standardizing variables with mean and pooled standardization of combined patients and caregivers [61].

### Ethical consideration

This study was approved by four Institutional Review Boards of the University affiliated hospitals (GAIRB 2019–187, HC19QEDI0070, HYUH 2019-12-017-008 and CNUH 2021–055). Patients with HF and their caregivers were screened and recruited during their hospital visits. After a patient and his/her caregiver signed a written informed consent form, trained research nurses at each hospital conducted face-to-face interviews to collect data on variables of interest in this study adhering to the research protocol to assure of data quality and avoid any biases. All the research processes were performed according to the principles of the Declaration of Helsinki [62].

## Results

### Sample characteristics of the patient-caregiver dyads

Fifty-five dyads (patients: 40.0% female, mean age = 65.4 ±14.6 years; caregivers: 96.4% female, mean age = 52.3 ± 12.6 years) participated in this study (Table 1). Regarding patient clinical

---

**Table 1. Demographic and disease-related characteristics of patients and caregivers (N = 55 dyads).**

| | Patients | Caregivers |
|---|---|---|
| | n (%) or Mean ± SD | n (%) or Mean ± SD |
| Age | 65.4±14.6 | 52.3±12.6 |
| Sex (female) | 22 (40.0) | 53 (96.4) |
| Marital (married) | 41 (74.5) | 45 (81.8) |
| Education | | |
| < High school | 28 (50.9) | 8 (14.5) |
| High school | 13 (23.6) | 30 (54.5) |
| College and higher | 14 (25.5) | 17 (30.9) |
| NYHA (I/II) | 37 (67.3) | |
| Etiology | | |
| Hypertension | 12 (21.8) | |
| Ischemic | 24 (43.6) | |
| Cardiomyopathy | 5 (9.1) | |
| Valvular heart disease | 2 (3.6) | |
| Alcoholic | 2 (3.6) | |
| Arrhythmia | 2 (3.6) | |
| Idiopathic | 8 (14.5) | |
| LVEF | 40.7±13.3 | |
| <40 | 28 (50.9) | |
| ≥40 | 27 (49.1) | |
| Relationship | | |
| Spouse | | 30 (54.5) |
| Children | | 17 (30.9) |
| Siblings | | 2 (3.6) |
| Parent | | 1 (1.8) |
| Grandchild | | 3 (5.5) |
| Missing | | 2 (3.6) |
| Disease (yes) | | 28 (50.9) |

*Note*. LVEF = left ventricular ejection fraction assessed by Echo. NYHA = New York Heart Association. SD = standard deviation.

characteristics, approximately two-thirds of patients (67.3%) were asymptomatic (NYHA functional class I) or mildly symptomatic (NYHA class II). The mean left ventricular ejection fraction was 40.7% (±13.3). Slightly less than half of the patients (43.6%) had an ischemic origin of HF. The patient-caregiver relationships were spouses (54.5%), adult children (30.9%), and other informal family caregivers (10.9%).

## Comparisons of physical function, depressive symptoms, patient self-care, and caregiver contribution to patient self-care

In comparisons of physical function and depressive symptoms in patient-caregiver dyads (Table 2), physical function of patients with HF was significantly poorer than that of caregivers (33.5 vs. 67.2, t = -9.70, $p < .001$). Depressive symptoms of patients were also significantly severer than those of caregivers (8.9 vs. 2.9, t = -6.25, $p < .001$).

Patient self-care (maintenance M ± SD = 52.7 ± 20.2; management = 46.5 ± 18.0) and CCPS (maintenance = 45.8 ± 19.6; management = 33.4 ± 23.7) were poor (Table 2). The scores of self-care management in patients and CCPS management in caregivers were significantly

**Table 2. Differences in physical function, depressive symptoms, self-care maintenance, and self-care management between patients with heart failure and their caregivers (N = 55 dyads).**

| | Patients | | Caregivers | | t or U or $x^2$ | p |
|---|---|---|---|---|---|---|
| | n (%) or Mean ± SD | Range | n (%) or Mean ± SD | Range | | |
| Physical function | 33.5±21.8 | 0–76.8 | 67.2±13.7 | 29.8–76.8 | -9.70 | < .001 |
| Depressive symptoms | 8.9±6.3 | 0.0–24.0 | 2.9±3.3 | 0.0–17.0 | -6.25 | < .001 |
| Self-care maintenance‡ | 52.7±20.2 | 7.5–92.5 | 45.8±19.6 | 3.3–90.0 | -1.83† | .067 |
| <70 (inadequate level) | 44 (80.0) | | 48 (87.3) | | 1.06 | .440 |
| ≥70 (adequate level) | 11 (20.0) | | 7 (12.7) | | | |
| Self-care management§ | 46.5±18.0 | 15.6–100.0 | 33.4±23.7 | 0.0–100.0 | -3.47† | < .001 |
| <70 (inadequate level) | 49 (89.1) | | 52 (94.5) | | 1.09 | .489 |
| ≥70 (adequate level) | 6 (10.9) | | 3 (5.5) | | | |

*Note.* †Mann-Whitney U

‡ Self-care maintenance in patients and contribution to self-care maintenance in caregivers

§ Self-care management in patients and contribution to self-care management in caregivers

SD = standard deviation.

different in patient-caregiver dyads (46.5 vs. 33.4, $p < .001$), but not the scores of patient self-care maintenance and caregiver contribution to self-care maintenance (52.7 vs. 45.8, $p = .067$). Using a cut-point of 70 or greater, small percentages of dyads demonstrated self-care adequacy. In patients, 20.0% and 10.9% showed self-care adequacy in the maintenance and management, respectively. In caregivers, 12.7% and 5.5% showed CCPS adequacy in the maintenance and management, respectively.

## Actor and partner effects of physical function and depressive symptoms for patient self-care and caregiver contribution to patient self-care

Fig 1 illustrates significant actor-partner relationships of physical function and depressive symptoms to patient self-care and CCPS. There were significant actor effects of patient and caregiver physical function on patient self-care management and CCPS management, respectively (Table 3). Better patient physical function was related to better patient self-care management (b = 0.30, $p = .005$, actor effect), and better caregiver physical function was related to better CCPS management (b = 0.54, $p = .015$, actor effect) (Table 3). In addition, there was a significant partner effect of patient physical function on CCPS maintenance. Better patient

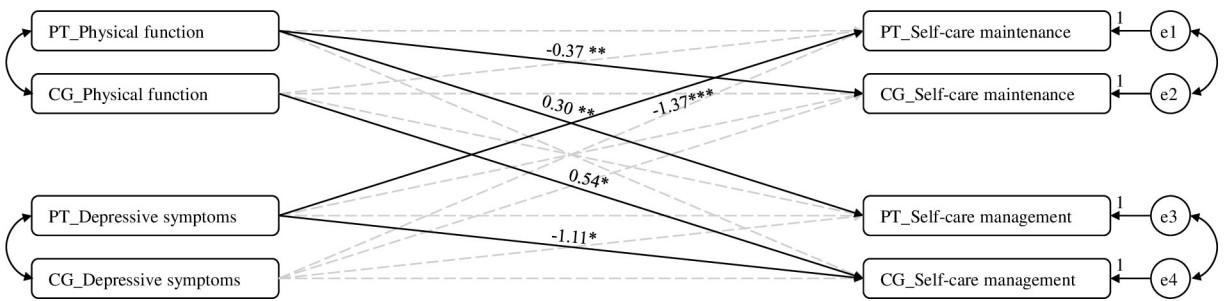

**Fig 1. Relationships of physical function and depressive symptoms to self-care.** *Note.* * $p < .05$, ** $p < .01$, *** $p < .001$. PT, patients; CG, caregivers; e1, error term of patients' self-care maintenance; e2, error term of caregivers' contribution to self-care maintenance; e3, error term of patients' self-care management; e4, error term of caregivers' contribution to self-care management. Continuous lines indicate significant relationships, and dotted lines indicate non-significant relationships.

**Table 3. Actor-Partner Interdependence Model parameter estimates for self-care maintenance and management.**

|  | Self-Care Maintenance[†] | | | | Self-Care Management[‡] | | | |
|---|---|---|---|---|---|---|---|---|
|  | b | beta | p | SE | b | beta | p | SE |
| Physical function | | | | | | | | |
| Actor effects | | | | | | | | |
| Patients | -0.07 | -0.05 | .548 | .11 | 0.30 | 0.30 | .005 | .11 |
| Caregivers | -0.24 | -0.17 | .190 | .18 | 0.54 | 0.48 | .015 | .22 |
| Partner effects | | | | | | | | |
| Patients on caregivers | -0.37 | -0.38 | .001 | .11 | -0.22 | -0.19 | .121 | .14 |
| Caregivers on patients | 0.27 | 0.21 | .136 | .18 | -0.02 | -0.01 | .926 | .17 |
| Depressive symptoms | | | | | | | | |
| Actor effects | | | | | | | | |
| Patients | -1.37 | -1.34 | < .001 | .39 | -0.22 | -0.22 | .536 | .36 |
| Caregivers | 0.22 | 0.26 | .770 | .74 | 0.01 | -0.05 | .989 | .90 |
| Partner effects | | | | | | | | |
| Patients on caregivers | -0.52 | -0.53 | .186 | .39 | -1.11 | -1.05 | .019 | .47 |
| Caregivers on patients | -0.19 | -0.24 | .799 | .74 | -0.11 | -0.11 | .869 | .68 |

*Note.*

[†]Self-care maintenance in patients and contribution to self-care maintenance in caregivers

[‡]Self-care management in patients and contribution to self-care management in caregivers

SE = standard error. b = unstandardized coefficient. beta = standardized coefficient.

physical function was related to poorer CCPS maintenance (b = -0.37, $p$ = .001) (partner effect). There was a significant actor effect of patient depressive symptoms on patient self-care maintenance and a partner effect on CCPS management. Severer depressive symptoms were related to poorer patient self-care maintenance (b = -1.37, $p$ < .001; actor effect) and poorer CCPS management (b = -1.11, $p$ = .019; partner effect).

## Discussion

To the best of our knowledge, we initially examined whether physical function and depressive symptoms of patient-caregiver dyads were related to patient self-care maintenance and management and CCPS maintenance and management in HF. The findings of this study partially supported the hypotheses suggested. In this study, actor-partner effects of physical function and depressive symptoms of dyads for self-care differed by self-care types. Actor effects in the relationships of better physical function to better patient self-care management and CCPS management were found, while a partner effect in the relationship of better patient physical function to poorer CCPS maintenance was found. An actor effect in the relationship of severer depressive symptoms to poorer patient self-care maintenance and a partner effect in the relationship of severer depressive symptoms to poorer CCPS management were found. However, caregiver physical function and depressive symptoms were not related to patient self-care maintenance or management (i.e., partner effects). The overall findings of this study suggest that clinicians and researchers need to consider patient-caregiver dyads and their physical and psychological conditions to improve patient self-care and CCPS.

According a recent meta-analysis mostly from the western countries [63], self-care in patients with HF was poor in all three self-care dimensions (maintenance, management and confidence). Adequate self-care maintenance (≥70 out of 100) was found only in 4 out of 36 studies, and adequate self-care management (≥70) was found in none of 35 studies. Patients

from Eastern Asian countries, including Korea, China, and Taiwan, also demonstrated poor self-care with mean scores of patient self-care maintenance ranging from 39.7 (China) to 66.0 (Taiwan) and patient self-care management ranging from 29.7 (Taiwan) to 63.1 (Taiwan) [63]. In this study, patient self-care was also very poor. The mean scores of patient self-care maintenance and management were 52.7 and 46.5, respectively. Using the cut-point of 70, only 20.0% and 10.9% of patients showed adequate patient self-care maintenance and management, respectively. Caregivers can contribute to patient self-care by helping their patients maintain recommended behavioral recommendations and perform management strategies when symptoms occur [13, 16]. However, CCPS maintenance (55.9–70.0 out of 100) and management (55.3–67.6 out of 100) were also poor in Italian caregivers of patients with HF [52, 53]. Caregiver contribution to patient self-care has not been frequently examined in East Asian countries. In China [64], the mean scores of CCPS maintenance and management were lower than those in Italy. In this study, the mean scores of CCPS maintenance (45.8) and management (33.4) were even slightly lower than those in China. Using the cut-point of 70, only 12.7% and 5.5% of caregivers showed adequate CCPS maintenance and management, respectively. The overall findings in the prior studies and in this study strongly demonstrate the needs for improvement in patient self-care and CCPS.

Poor self-care can be attributed to patients' and/or their caregivers' depressive symptoms or impaired physical function [30, 34, 38, 65]. Depressive symptoms are very common in patients with HF in East Asians and Western countries, ranged from 32% to 46% [21, 22, 66]. In this study, the prevalence of depressive symptoms in patients with HF (45.5%) was similar to that in the prior studies. Depressive symptoms are also common in caregivers of patients with HF in Eastern and Western countries, ranging from 6% to 64% [23–25]. The prevalence of depressive symptoms in both patients and their caregivers in patient-caregiver dyads was 21% [26]. In this study, the prevalence of depressive symptoms in caregivers (3.6%) was similar to the lowest prevalence in other studies. On the other hand, impaired physical function in patients with HF and their caregivers has been reported in prior studies [35–37, 42, 43]. In this study, physical function score in patients was 33.5 out of 79 ($\geq$ 46: no functional limitation), indicating considerable impairment of physical function, while physical function score in caregivers (67.2) was considerably better than that in patients and indicated no functional limitation in caregivers. Thus, overall, our findings in conjunction with prior studies demonstrate high prevalence of depressive symptoms and impaired physical function in patients, while relatively varied prevalence in caregivers depending on studies. However, in prior studies, the prevalence of both depressive symptoms and physical function in patient-caregiver dyads has not been frequently examined.

In this study, each of physical function and depressive symptoms in patients had both actor and partner effects on one type of patient self-care and one type of CCPS. These findings indicate that the impaired physical and psychological conditions of patients adversely affect not only patient self-care but also CCPS. Contrastingly, caregiver physical function had only the actor effect on CCPS, while caregiver depressive symptoms showed no actor or partner effects on either patient self-care or CCPS. These findings provide valuable insights regarding the dyadic relationships of physical function and depressive symptoms to patient self-care or CCPS that have not been tested fully in HF, especially using APIM. Previous studies [30, 38] examined and supported the relationships of physical function and/or depressive symptoms in patients with patient self-care, consistent with our study findings. However, the relationships of physical function and depressive symptoms in patients to CCPS have been rarely examined in HF. Nevertheless, the study findings suggest their effects on CCPS maintenance and management. Considering the significant relationships between CCPS and patient self-care [52], these findings emphasize the importance of improvements in physical function and depressive

symptoms in patients with HF to improve patient self-care. On the other hand, the relationships of physical symptoms and depressive symptoms in caregivers to patient self-care and CCPS have been rarely examined among patient-caregiver dyad in HF. The finding of this study demonstrated the importance of caregiver physical function in CCPS maintenance. One possible reason for no other actor and partner effects of caregiver physical function and depressive symptoms on patient self-care and CCPS may be relatively good physical function and a very low prevalence of depressive symptoms in this caregiver sample. In other studies, impaired physical function and high prevalence of depressive symptoms in caregivers of patients with HF have been reported [23–25, 42, 43]. Therefore, further studies are needed to examine the comprehensive relationships of physical function and depressive symptoms to patients' self-care and CCPS in patient-caregiver dyads in larger sample studies with appropriate prevalence of impaired physical function and depressive symptoms.

Dyadic perspectives of HF self-care and influential factors for self-care have received increased attention over last decade, while evidence remains lacking. Our results indicate several implications for clinical practice and research. The findings of this study imply that assessment of physical and psychological conditions of both patients and caregivers are critically important to improve patient self-care and caregiver contribution in clinical practice. Poor self-care and CCPS in Korean patients and their caregivers also warrant educational support to promote their understanding of the self-care concept and the importance of caregiver assistance in the self-care process, especially in caregivers who showed substantially poor CCPS in clinical practice.

The findings of this study suggest the directions for practice and future studies. In this study, the prevalence of inadequate patient self-care, inadequate CCPS, impaired physical function in patients, and depressive symptoms in patients was high. In addition, the findings also suggest the actor and partner effects of physical function and depressive symptoms in patients on patient self-care and CCPS. Furthermore, physical function in caregivers also showed the actor effect on CCPS management. All these findings suggest the needs for improvements in physical function in patients and in caregivers and depressive symptoms in patients. Thus, clinicians need to assess these variables and develop and deliver appropriate interventions to patient-caregiver dyads in order to improve these variables, and, in turn, to improve patient self-care. The intervention effects on both physical function and depressive symptoms in patient-caregiver dyad have not been frequently tested in HF. A mobile health intervention study was provided to both patients and caregivers, and improved patients' self-care and emotional status, but the effects on caregivers have not been examined [67]. An education and psychosocial support intervention was also provided to both patients with HF and their caregivers, but did not affect depressive symptoms in patients and caregivers, self-care in patients, and caregiver burden in caregivers [68]. Thus, researchers and clinicians need to develop more effective interventions to improve physical function and depressive symptoms, and, in turn, CCPS and patient self-care in patient-caregiver dyads. In addition, further research is also needed to test the hypothesized actor and partner relationships in patient-caregiver dyads using more rigorous study designs, such as longitudinal study designs and larger samples with higher prevalence of impaired physical function and depressive symptoms, especially in caregivers.

This study had several limitations. Although our sample for the dyadic analysis was justified with the adequate power, the validation of the results is warranted in a larger sample study with appropriate prevalence of impaired physical function and depressive symptoms in caregivers. Patients and their caregivers were recruited using a convenience sampling method, limiting the generalization of the study findings. Further, the nature of the cross-sectional design in this study remains unknown whether improvements in physical and psychological

conditions of patient-caregiver dyads will lead to improvements in the dyadic patient self-care and CCPS. Moreover, the study findings may be influenced by cultural factors impacting patient self-care and CCPS, limiting their direct applicability to other cultural or geographical contexts.

The study findings propose several approaches for future research. Firstly, exploring larger samples with random sampling can enhance the robustness of conclusions. Secondly, employing a longitudinal design to examine changes over time and establish causality is recommended. Thirdly, considering the incorporation of various characteristics, including socioeconomic status, education, and social support, can contribute to gaining a more comprehensive understanding of factors associated with self-care.

## Conclusions

The findings of this study partially supported the hypothesized actor and partner effects of physical function and depressive symptoms on patient self-care and CCPS in patient-caregiver dyads. Impaired physical function and depressive symptoms in patients showed both the actor and partner effects on poor self-care in patients and also poor CCPS in caregivers. The partner effect of physical function in patients and the actor and partner effects of depressive symptoms in caregivers and in patients, respectively, were not supported. However, the relatively low prevalence of impaired physical function and depressive symptoms in this caregiver sample warrants further studies to examine the hypothesized actor and partner effects using caregiver samples with relatively high prevalence of impaired physical function and depressive symptoms.

## Supporting information

**S1 Dataset. Study dataset.**
(ZIP)

## Author Contributions

**Conceptualization:** JinShil Kim.

**Formal analysis:** Minjeong An.

**Funding acquisition:** JinShil Kim.

**Investigation:** JinShil Kim, Kye Hun Kim, Mi-Seung Shin, Minjeong An.

**Project administration:** JinShil Kim, Kye Hun Kim, Mi-Seung Shin, KyungAh Cho.

**Resources:** KyungAh Cho.

**Supervision:** JinShil Kim.

**Validation:** Seongkum Heo, Jung-Ah Lee, KyungAh Cho.

**Writing – original draft:** JinShil Kim, Seongkum Heo, Minjeong An.

**Writing – review & editing:** JinShil Kim, Kye Hun Kim, Mi-Seung Shin, Seongkum Heo, Jung-Ah Lee, KyungAh Cho.

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
