## [Decision Letter · Decision Letter 0]

9 Jan 2024

PONE-D-23-25156Evaluating Dyadic Factors Associated with Self-Care in Patients with Heart Failure and Their Family Caregivers: Using an Actor-Partner Interdependence ModelPLOS ONE

Dear Dr. An,

Thank you for submitting your manuscript to PLOS ONE. After careful consideration, we feel that it has merit but does not fully meet PLOS ONE’s publication criteria as it currently stands. Therefore, we invite you to submit a revised version of the manuscript that addresses the points raised during the review process.

We look forward to receiving your revised manuscript.

Kind regards,

Shigao Huang

Academic Editor

PLOS ONE

Journal Requirements:

Reviewers' comments:

Reviewer's Responses to Questions

**Comments to the Author**

1. Is the manuscript technically sound, and do the data support the conclusions?

Reviewer #1: Yes

Reviewer #2: Yes

2. Has the statistical analysis been performed appropriately and rigorously? 

Reviewer #1: I Don't Know

Reviewer #2: Yes

3. Have the authors made all data underlying the findings in their manuscript fully available?

Reviewer #1: Yes

Reviewer #2: Yes

4. Is the manuscript presented in an intelligible fashion and written in standard English?

Reviewer #1: Yes

Reviewer #2: Yes

5. Review Comments to the Author

Reviewer #1: In this article entitled “Evaluating dyadic factors associated with self-care in patients with heart failure and their family caregivers: using an Actor-Partner Interdependence Model” the authors analyze how relationships of patient-caregiver physical function and depressive symptoms to the self-care behavior of the disease. A study highlighting the need to assess the physical and psychological conditions of both patients and caregivers, and the importance of caregiver assistance in self-care. The manuscript addresses an important and interesting topic, especially that chronic HF is increasing in prevalence with an ageing population. However, I have a few comments on some issues that require more attention.

Introduction:

• In addition to the recent review mentioned in the authors' review, I found a more recent review: Uchmanowicz I, et al. Heart Failure Care: Testing Dyadic Dynamics Using the Actor-Partner Interdependence Model (APIM)-A Scoping Review. Int J Environ Res Public Health. 2022 Feb 9;19(4):1919. doi: 10.3390/ijerph19041919. PMID: 35206131; PMCID: PMC8871794, and it seems reasonable to include it as well, particularly as it talks about the emotional components of HF management within the framework of APIM.

Discussion

• the discussion part could be a little bit shorter and easier to follow

• In the discussion section, I would recommend avoiding figures from other authors' studies such as," In China, the mean scores of CCPS maintenance and management were 52.4 and 55.6 [63], which are similar or somewhat lower than those in Italy."

Reviewer #2: This study used a cross-sectional, correlational design to investigate the physical function and depressive symptom status of patients with heart failure (HF) and their caregivers as dyads, examining their associations with patient self-care and caregiver contribution to patient self-care (CCPS). The study collected data from four university-affiliated hospitals in South Korea between June 12, 2019, and October 29, 2021.

The study focused on patient-caregiver dyads, which provides a more comprehensive understanding of the interactions between patients and their caregivers, as opposed to solely studying patients or caregivers in isolation.

Heart failure is a significant health concern, and understanding the factors affecting self-care and caregiver contributions is essential for improving the quality of care for patients with HF.

The study employed well-established tools, such as the Revised Self-Care of Heart Failure Index and the Patient Health Questionnaire-9, to assess self-care and depressive symptoms. This enhances the validity of the findings.

The study was conducted following ethical guidelines, including obtaining informed consent from participants and approval from Institutional Review Boards, ensuring the ethical conduct of research.

The study used a convenience sampling method, which may limit the generalizability of the findings to a broader population. Convenience samples might not represent the entire population of patients with HF and their caregivers.

The study's cross-sectional design limits its ability to establish causal relationships. Longitudinal studies would be more suitable for examining changes over time and identifying causality.You should adress it in the paper as the study limitations.

While the sample size was deemed adequate for the power analysis, larger samples could enhance the robustness of the findings, especially given the complexity of the dyadic relationships being examined.

The study was conducted in South Korea, which may have cultural and regional nuances affecting self-care and caregiver contributions. The findings may not be directly transferable to other cultural or geographical contexts.

The study primarily focused on physical function and depressive symptoms as factors influencing self-care. Other variables, such as socioeconomic status, education, and social support, may also play significant roles but were not explored.

In summary, this study offers valuable insights into the relationships between physical function, depressive symptoms, self-care, and caregiver contributions in patient-caregiver dyads dealing with heart failure. However, it has limitations, and future research should aim for larger, more diverse samples and consider additional factors that may impact self-care and caregiving.

6. PLOS authors have the option to publish the peer review history of their article (what does this mean?). If published, this will include your full peer review and any attached files.

Reviewer #1: No

Reviewer #2: **Yes: **Izabella Uchmanowicz

---

## [Author Response · Author response to Decision Letter 0]

4 Mar 2024

Dear Editor:

We would like to thank you for your thoughtful comments and recommendations. We have made the following changes to the manuscript to address your concerns and revised the paper, paying particular attention to language. We obtained additional editing support from a scientific editing company. The revised sections of the manuscript are indicated in RED.

Please see the respond to reviewers file attached.

Thank you again for your constructive feedback.

Sincerely,

Authors

---

## [Decision Letter · Decision Letter 1]

7 Jun 2024

PONE-D-23-25156R1Evaluating Dyadic Factors Associated with Self-Care in Patients with Heart Failure and Their Family Caregivers: Using an Actor-Partner Interdependence ModelPLOS ONE

Dear Dr. An,

Thank you for submitting your manuscript to PLOS ONE. After careful consideration, we feel that it has merit but does not fully meet PLOS ONE’s publication criteria as it currently stands. Therefore, we invite you to submit a revised version of the manuscript that addresses the points raised during the review process.

We look forward to receiving your revised manuscript.

Kind regards,

Shigao Huang

Academic Editor

PLOS ONE

Journal Requirements:

Reviewers' comments:

Reviewer's Responses to Questions

**Comments to the Author**

1. If the authors have adequately addressed your comments raised in a previous round of review and you feel that this manuscript is now acceptable for publication, you may indicate that here to bypass the “Comments to the Author” section, enter your conflict of interest statement in the “Confidential to Editor” section, and submit your "Accept" recommendation.

Reviewer #2: All comments have been addressed

2. Is the manuscript technically sound, and do the data support the conclusions?

Reviewer #2: (No Response)

3. Has the statistical analysis been performed appropriately and rigorously? 

Reviewer #2: Yes

4. Have the authors made all data underlying the findings in their manuscript fully available?

Reviewer #2: Yes

5. Is the manuscript presented in an intelligible fashion and written in standard English?

Reviewer #2: Yes

6. Review Comments to the Author

Reviewer #2: Since the title is built on this manuscript it is worth to add the citation Uchmanowicz I, Faulkner KM, Vellone E, Siennicka A, Szczepanowski R, Olchowska-Kotala A. Heart Failure Care: Testing Dyadic Dynamics Using the Actor-Partner Interdependence Model (APIM)-A Scoping Review. Int J Environ Res Public Health. 2022 Feb 9;19(4):1919. doi: 10.3390/ijerph19041919. PMID: 35206131; PMCID: PMC8871794.

7. PLOS authors have the option to publish the peer review history of their article (what does this mean?). If published, this will include your full peer review and any attached files.

Reviewer #2: **Yes: **Izabella Uchmanowicz

---

## [Author Response · Author response to Decision Letter 1]

22 Jul 2024

July 22, 2024

Shigao Huang

Academic Editor

Plos One

Dear Editor:

Thank you very much for your feedback regarding the reference list in our manuscript titled “Evaluating Dyadic Factors Associated with Self-Care in Patients with Heart Failure and Their Family Caregivers: Using an Actor-Partner Interdependence Model.” We have thoroughly reviewed our references and incorporated the necessary adjustments. The revised sections of the manuscript are indicated in RED.

Please see the response to reviewers file attached.

Thank you again for your constructive feedback.

Sincerely,

Authors

---

## [Editor Report · Decision Letter 2]

25 Jul 2024

Evaluating Dyadic Factors Associated with Self-Care in Patients with Heart Failure and Their Family Caregivers: Using an Actor-Partner Interdependence Model

PONE-D-23-25156R2

Dear Dr. An,

We’re pleased to inform you that your manuscript has been judged scientifically suitable for publication and will be formally accepted for publication once it meets all outstanding technical requirements.

Kind regards,

Shigao Huang

Academic Editor

PLOS ONE
---

## [Editor Report · Acceptance letter]

2 Aug 2024

PONE-D-23-25156R2 

PLOS ONE

Dear Dr. An, 

I'm pleased to inform you that your manuscript has been deemed suitable for publication in PLOS ONE. Congratulations! Your manuscript is now being handed over to our production team.

Kind regards, 

on behalf of

Dr. Shigao Huang 

Academic Editor

PLOS ONE